# Numerical and Experimental Investigations of Cold-Sprayed Basalt Fiber-Reinforced Metal Matrix Composite Coating

**DOI:** 10.3390/ma16051862

**Published:** 2023-02-24

**Authors:** Sihan Liang, Yingying Wang, Bernard Normand, Yingchun Xie, Junlei Tang, Hailong Zhang, Bing Lin, Hongpeng Zheng

**Affiliations:** 1School of Chemistry and Chemical Engineering, Southwest Petroleum University, Chengdu 610500, China; 2Key Laboratory of Optoelectronic Chemical Materials and Devices (Ministry of Education), Jianghan University, Wuhan 430056, China; 3The Key Lab of Guangdong for Modern Surface Engineering Technology, National Engineering Laboratory for Modern Materials Surface Engineering Technology, Institute of New Materials, Guangdong Academy of Sciences, Guangzhou 510651, China; 4Institut National des Sciences Appliquées de Lyon | INSA Lyon · Laboratory of Materials: Engineering and Science (MATEIS-UMR 5510), Université de Lyon, Bat L. de Vinci, 21 Avenue Jean Capelle, Villeurbanne CEDEX, 69621 Lyon, France

**Keywords:** cold spraying, co-deposition mechanism, fiber-reinforced composite coating, numerical simulation

## Abstract

The aluminum-basalt fiber composite coating was prepared for the first time with basalt fiber as the spraying material by cold-spraying technology. Hybrid deposition behavior was studied by numerical simulation based on Fluent and ABAQUS. The microstructure of the composite coating was observed on the as-sprayed, cross-sectional, and fracture surfaces by SEM, focusing on the deposited morphology of the reinforcing phase basalt fibers in the coating, the distribution of basalt fibers, and the interaction between basalt fibers and metallic aluminum. The results show that there are four main morphologies of the basalt fiber-reinforced phase, i.e., transverse cracking, brittle fracture, deformation, and bending in the coating. At the same time, there are two modes of contact between aluminum and basalt fibers. Firstly, the thermally softened aluminum envelops the basalt fibers, forming a seamless connection. Secondly, the aluminum that has not undergone the softening effect creates a closed space, with the basalt fibers securely trapped within it. Moreover, the Rockwell hardness test and the friction-wear test were conducted on Al–basalt fiber composite coating, and the results showed that the composite coating has high wear resistance and high hardness.

## 1. Introduction

As a new material deposition technique [1], cold spraying (CS) has been widely used in the preparation of non-metallic reinforced-phase composite coatings due to its advantage of avoiding the disadvantages of traditional thermal spraying such as oxidation and phase change [2,3,4,5]. Among these, ceramic materials have become popular for research due to their significant advantages in enhancing the wear resistance and hardness of composite coatings. Common ceramic reinforcing phases include Al_2_O_3_ [6], SiO_2_ [7], ${\mathrm{ZrO}}_{2}$ [8], TiC [9], SiC [10], and WC [11].

However, ceramics are brittle materials with poor ductility. To overcome this problem, softer and more ductile metals are often used as auxiliary adhesives. A lot of research has shown that ceramics of different morphologies can be used as reinforcement phases to improve the performance of coatings [12,13,14,15]. For example, Fernandez [16] prepared Al-Al_2_O_3_ composite coatings by mixing spherical ceramic Al_2_O_3_ and Al and found that the plastic deformation of Al particles helps the deposition of ceramic particles. Sansoucy et al. [17] used irregularly shaped SiC as a reinforcement phase to prepare Al-SiC composite coatings, and found that the addition of SiC significantly improved the hardness and bonding strength of the coatings. Suo et al. [18] used porous WC-17Co particles and Al metal for mixed deposition, and found that the brittleness of WC-17Co affected the bonding force between the coating and the substrate. Yin et al. [19] successfully deposited the diamond in its original phase on an Al substrate by using Cu powder to wrap the diamond in a cubic shape, which improved the wear resistance of the coating. At the same time, the shape and size of ceramics have a great influence on the as-deposit morphology of ceramics and the properties of composite coatings [14,15,20]. Jodoin et al [16]. found that the breaking of SiC in the mixed deposition of SiC and Al led to an increase in the porosity of composite coatings. Chen et al. [21] prepared composite coatings of Cu and Al_2_O_3_, SiC, and WC ceramics, respectively, to explore the effect of different ceramic materials on the wear resistance and thermal conductivity of coatings and found that Al_2_O_3_- and SiC-enhanced composite coatings had better wear resistance and thermal conductivity. In order to improve the efficiency of ceramic deposition, Yin et al. [22] prepared high wear-resistant coatings by mixing WC-Co and Ni powder. During the deposition process, it was found that the structure fracture of the porous reinforcing phase material WC-Co largely preserved the WC-Co content in the coating, increasing its wear resistance. To sum up, the shape and size of ceramics play a crucial role in the performance regulation of composite coatings. Hence, it is imperative to closely monitor these aspects to attain composite coatings with exceptional performance [14,15,16,17,18,20,22].

Fiber-reinforced composites are a widely used type of composite material. To date, in the field of ceramic spraying (CS), fibers have only been studied as substrates that need to be metalized [23,24,25], while the use of fibers as a raw powder for CS has not been reported. Due to its unique fiber-like morphology and large aspect ratio, fiber reinforcements will exhibit different deposition behavior in CS compared to particle reinforcements. In this study, short chopped basalt fibers were used as the reinforcement phase and the deposition behavior on metal surfaces was studied through both simulation calculation and experimental methods. The co-deposition process of aluminum and basalt fibers was studied using the Coupled Euler–Lagrange (CEL) method and the ABAQUS Display dynamic analysis module. The coating morphologies observed through electron microscopy were compared with the calculation results, confirming four deposition forms of the basalt fibers. In addition, hardness and wear resistance tests showed that the addition of basalt fibers improved the performance of the coating.

## 2. Numerical Modeling

### 2.1. Finite Element Method

Numerical simulation was carried out using the ABAQUS/Explicit (Version: 2021), and the impact behavior between the Al–basalt mixed particles and Al substrate was studied using a three-dimensional (3D) model. Due to the different brittle fractures of non-metallic materials in CS and the deformation behavior of metallic materials, the metal-ceramic CS has been simulated by using the coupled Voronoi and cohesive element methods in some previous studies [26]. In this study, a similar approach has been adopted using the Lagrangian method to model the particles and insert zero-thickness cohesive elements into the 3D model of the basalt fiber. The substrate is modeled using the CEL method, and three different cases are considered: (i) deposition of single Al powder particle, as shown in Figure 1a,b; (ii) deposition of single-particle basalt fiber and subsequent impact of single Al particle, as shown in Figure 1c,d; (iii) deposition of 15 vol.% and 30 vol.% basalt fiber-Al mixed particles, as shown in Figure 1e,f.

Using the CREO (Version:7.0.0.0) 3D modeling software, Al particles, basalt fiber, and substrate have been modeled in the three cases. A mesh convergence study was performed to determine the optimal mesh size of 1/100 dp. The particles were meshed using C3D8RT with deformation and enhanced hourglass control, and the substrate was meshed using an eight-node linear Euler element (EC3D8R) with reduced integration points. To reduce the computational cost, a transitional mesh was used for meshing the substrate. A larger mesh was used outside the impact position, and to avoid the numerical stability problem caused by the stress wave reflection at the model boundary, the overall size of the substrate was taken as 10 times the particle diameter. A total of 10 groups of 100 Al particles with $d_{p}=$ 10–100 μm particle diameter were created using a Python script and then the Al particles and 15 and 30 vol.% basalt fibers were randomly distributed in the specified rectangular area, as shown in Figure 1e. Symmetrical boundary conditions were used for the symmetry plane of the model, and boundary condition $V_{i}=0$ was defined for surfaces other than the impact surface of the substrate, where i represents the coordinate axis perpendicular to the surface.

The deposition conditions of single particles of Al and basalt fiber and multiple particles of Al–basalt fiber with a specific mixing ratio were determined by the fluid mechanic’s method based on Ansys Fluent (Version: 2022 R1) software. Modeling according to the specific dimensions of the Laval nozzle used. The two-dimensional (2D) nozzle was planarly meshed using ICEM, and the near-wall mesh was refined to ensure accurate results. For simulating the fluid–structure interaction in Ansys Fluent, the first step was steady-state calculation, and the entire computational domain of the nozzle was initialized by full multigrid (FMG) suitable for the fluid mechanic’s calculation method. The second step was to set the gas type as N_2_, the pressure as 5 MPa, and the temperature as 380 °C according to the actual spraying parameters.

### 2.2. Numerical Model

Assuming that the Al powder has a linear elastic response, the Johnson–Cook (JC) plastic model is used to model its constitutive relation, which includes the effects of strain hardening, strain rate hardening, and thermal softening on yield stress. The model [27] is expressed as follows:(1)$$\sigma \left(\varepsilon^{P},{\dot{\varepsilon }}^{P},T\right)=\left[A+B{\left(\varepsilon^{P}\right)}^{n}\right]\left[1+Cln\left(\frac{{\dot{\varepsilon }}^{P}}{{\dot{\varepsilon_{r}}}^{P}}\right)\right]\left[1-T^{*}{}^{m}\right]$$
(2)$$T^{*}=\{\begin{array}{ccc} \frac{T-T_{r}}{T_{m}-T_{r}} & if & T_{r}\leq T\leq T_{m}\\ 1 & if & T<T_{r}\\ 0 & if & \;T\geq T_{m} \end{array}$$
where A, B, n, C, and m are material-dependent constants [28,29], $\sigma \left(\varepsilon^{P},{\dot{\varepsilon }}^{P},T\right)$ is the flow stress, ${\dot{\varepsilon }}^{P}$ is the strain rate, ${\dot{\varepsilon_{r}}}^{P}$ is the reference strain rate, T is the current temperature, $T_{r}$ is the reference temperature, and $T_{m}$ is the melting temperature of the material. In this section, the modified formula of Rohan et al. [30] is adopted. In the JC plastic model considering the plastic strain gradient, it is considered that
(3)$$\sigma_{JC}=\left[A+B\varepsilon^{n}\right]\left[1+Cln\frac{\dot{\varepsilon_{p}}}{\dot{\varepsilon_{0}}}{\left(\frac{\dot{\varepsilon_{p}}}{\dot{\varepsilon_{c}}}\right)}^{D}\right]\left[1-T^{*}{}^{m}\right]$$
where
(4)$$D=\{\begin{array}{ll} 0, & \dot{\varepsilon_{p}}<\dot{\varepsilon_{c}}\\ x, & \dot{\varepsilon_{p}}\geq \dot{\varepsilon_{c}} \end{array}\;and\;\dot{\varepsilon_{c}}=ys^{-1}$$

As previously explained by Assadi et al. [2], the deformation process is considered to be adiabatic. Therefore, the temperature rise caused by the deformation is not negligible. By coupling the constitutive equation with the thermal equation, the temperature rise caused by the deformation can be calculated as follows:(5)$$\nabla T\left(\varepsilon^{P},{\dot{\varepsilon }}^{P},T\right)=\frac{\beta }{\rho C_{P}}\;\int_{\varepsilon }^{\varepsilon^{P}}\sigma \left(\varepsilon^{P},{\dot{\varepsilon }}^{P},T\right)d\varepsilon^{P}$$
where β is the Quinney–Taylor heat fraction coefficient [31,32,33], ρ is the density of the material, and $C_{P}$ is the specific heat at a constant pressure. Several researchers have used the Taylor series method for polymers. In this study, the proportion of plastic energy converted to heat energy is designated as β=0.9, and the coefficient value has been provided in previous studies [34,35]. The basic material parameters of Al are listed in Table 1 [36]. The basic material parameters of basalt fibers are derived from monofilament tensile tests.

Numerical modeling was performed based on the experimentally measured dimensions of basalt fibers, where the diameter is 12 μm and the length is 60–100 μm. A three-dimensional model of basalt fibers was created by the method of the Voronoi diagram. This is a method that forms irregular polyhedra by randomly generating seed points inside the model that do not affect each other and then forming irregular polyhedra with the seed points as centers. Since a large number of generated irregular polyhedra are difficult to mesh, the model is simplified by adjusting the number of seed points in the basalt fiber model. Meanwhile, cohesive cells are inserted on the surface of the generated irregular polyhedra as potential crack expansion paths of basalt fibers. As shown in Figure 2a, multiple irregular polyhedra are generated to form basalt fibers by adjusting the seed points. These irregular polyhedra are not crystals, but only viscous elements are inserted on the surface of the irregular polyhedra as potential fracture expansion paths for basalt fibers.

The initiation and propagation of the basalt fiber cracks are investigated by the cohesive force finite element method, which is mainly governed by the bilinear viscous traction separation law, as shown in Figure 2c. This method can effectively describe the mechanical properties of crack propagation [37,38]. The mode of crack generation and propagation is described in Figure 2b,d. When the traction force at the interface T is maximized (T_max_), the stress in the material reaches its critical strength and damage occurs. Under the action of traction, the two surfaces inserted in the cohesive element extend in the normal and tangential directions of the traction force.

The fracture area under the tension separation curve is equal to the interfacial fracture energy ΓC, as shown in Equation (6). With the development of damage, the interfacial traction force linearly decays to zero, while the interfacial displacement δ approaches the maximum value. For brittle materials with extremely small plastic deformation before damage such as basalt fiber and ceramic the separation form of triangular traction is the most suitable. As shown in Equation (7), the traction separation model of crack extension can be represented by an elastic constitutive matrix that relates the normal stress and shear stress with the normal separation and shear separation displacements at the interface. In this study, the cohesive element inserted into an irregular polyhedron is defined as a viscous surface with zero thickness, where failure can only occur under the action of pure tension or shear stress. Therefore, in the interface separation shown in Equation (6), the relative displacement between a node on one surface and the corresponding projected point on a connected surface is along the contact normal and shear direction.
(6)$$\Gamma_{c}=\frac{1}{2}T_{max}\delta_{max}$$
(7)$$T=\left\{\begin{array}{c} T_{n}\\ T_{s}\\ T_{t} \end{array}\right\}=\left[\begin{array}{ccc} K_{nn} & K_{ns} & K_{nt}\\ K_{ns} & K_{ss} & K_{st}\\ K_{nt} & K_{st} & K_{tt} \end{array}\right]\left\{\begin{array}{c} \sigma_{n}\\ \sigma_{s}\\ \sigma_{t} \end{array}\right\}=K\delta$$
where the nominal traction stress vector T consists of three components ($T_{n},\;T_{s},\;T_{t}$), which represents the normal traction stress ($T_{n}$) and its two shear components ($T_{s},\;T_{t}$). The traction force is proportional to the corresponding displacement in the 3D axis (X is the relative axis *x*, *y*, *z*), and it is related to the normal and tangential stiffness ($K_{nn},\;K_{ss},\;K_{tt}$). $T_{max}$ represents the maximum traction force at the beginning of damage, and $\delta_{max}$ represents the maximum displacement under crack propagation. In Equation (6), the normal and tangential stiffness components are separate, i.e., pure normal separation does not produce cohesion in the shear direction and vice versa [39]. Turon et al. [35] suggested the following equation for calculating the stiffness of bonded surfaces:(8)$$K\geq \frac{\alpha E}{Heff}$$
where α = 50, and Heff is the thickness of the adjacent element. The basalt fiber used in this paper is assumed to be isotropic in terms of size and stiffness, i.e., $K_{nn}=K_{ss}=K_{tt}$. The cylindrical basalt fiber is modeled as a continuum with isotropic material to facilitate the simulation of fractures. The overall cohesive surface area has the properties of interactions, not cohesive units. Therefore, the surface traction force and separation of the basalt fiber can be expressed by this formula. According to the experimental data of basalt fiber monofilament tension, the monofilament tension strength of basalt fiber is obtained as $\sigma_{s}=$ 3773 MPa. The grid is taken as 1/25 of the model size, and the single grid length is 0.6 μm. After the above equation, the basic data related to the cohesive element of basalt fiber are listed in Table 2.

### 2.3. Simulations

The initial velocity and temperature of all particles were calculated based on the Fluent aerodynamics method at an initial condition of 5 MPa, 380 °C, and the temperature of the substrate was kept constant (298 K) in all numerical simulations. The particle deposition process was numerically simulated by the CEL method, which has the advantages of obvious particle-substrate boundary, good mesh adaptation, and high accuracy. In order to verify and explore the deformation ability of aluminum metal, the impact test of single-particle Al powder was established. Among them, the size of Al particles is 55 μm as the average value in the diameter range of 10–100 μm. To explore the applicability of the cohesive unit in the basalt fiber model, the single-particle basalt fiber impact test was conducted to investigate the effect of subsequent Al particles on the impact strength of basalt fibers. In a multi-particle mixed random spray, the Al particles had diameters ranging from 10–100 μm, and the number of particles of different diameters was determined using laser particle size analysis. The lengths of the basalt fibers were 60, 80, and 100 μm, and the relevant parameters were determined according to the specific mixing ratios. In the numerical simulations, it can be reasonably assumed that the initial temperature and velocity of each group of particles with different diameters, calculated by the Fluent aerodynamic method, can be used for subsequent numerical simulation calculations of the mixing deposition.

## 3. Experimental

In this study, we analyzed the cross-section of the composite coating by SEM (Model: German ZEISS GeminiSEM 300) and EDS (Model: OXFORD Xplore) and sprayed the sample with gold by sputtering coater (Model: Quorum SC7620) before the test. Pure aluminum powder (New Material Co., Ltd.: Guangdong, China) was used. Basalt fibers were provided by Aerospace Tuoda Basalt Delopment Co., Ltd. (Dazhou, Sichuan, China). Aluminum material was used as an auxiliary metal binder, The size of the Al particles was determined using laser diffraction analysis(Model: Better BT-9300SE). The size of Al powder used in the experiment was found to be 10–180 μm. Among them, more than 95% of the pure aluminum particles have a diameter of 10–100 um, and less than 5% of the pure aluminum particles have a diameter of 100–180 μm. Therefore, in the numerical modeling, we use pure aluminum particles with a diameter of 10–100 μm and an average particle size of 55 μm. The SEM images of the basalt fibers are shown in Figure 3. The diameter and length of the basalt fibers are 10–14 and 66–148 μm, respectively. The cumulative probability distribution of particle size can be estimated by the lognormal function [39] as follows:(9)$$f\left(d_{p}\right)=50\left[1+erf\left(\frac{ln\left(d_{p}\right)-ln\left({\bar{d}}_{p}\right)}{\sigma_{log}\sqrt{2}}\right)\right]\;$$
where ${\bar{d}}_{p}$ is the mass average diameter of powder particles, and $\sigma_{log}=0.36$. Based on a Python script, 100 Al particles were generated, and the particle diameter was $d_{p}=$ 10–100 mm.

The Al–basalt composite coating was prepared by using the cold-spraying equipment PCS1000 of Guangzhou Nonferrous Metals Research Institute. The nozzle had a circular section with an expansion ratio of 13.4. The nozzle throat diameter was 3 mm, and the lengths of the nozzle convergent section and divergence section were 80 and 110 mm, respectively. The inlet and outlet diameters were 26 and 11 mm, respectively. The relevant process parameters are shown in Table 3.

Chopped basalt fiber and soft aluminum metallic powder were used in the experiment, and the content of basalt fiber was 15 vol.% or 30 vol.%. The powder-feeding gas was N_2_; the initial pressure and temperature were 5 MPa and 380 °C, respectively. The spraying process is shown in Figure 4.

Through this experiment, 15 vol.% and 30 vol.% basalt-reinforced composite coatings were successfully prepared on Al substrate under the conditions of 5 MPa and 380 °C by the PCS1000 equipment. To investigate the wear resistance of the composite coating, the friction-wear test was performed on composite coatings with basalt contents of 15 vol.% and 30 vol.%. The relevant experimental data are shown in Table 4.

## 4. Results and Discussion

To study the deposition behavior of Al–basalt fiber on Al substrate, the deformation ability of Al powder was firstly examined through numerical simulation, and the impact behavior of a single Al particle was simulated. Secondly, the position of a single basalt particle after inserting a cohesive unit was examined to reveal the breaking behavior of a single basalt fiber. Finally, based on the numerical simulation results of single-particle deposition, 15 vol.%, and 30 vol.% basalts fiber-Al multi-particle deposition behavior was numerically investigated.

### 4.1. Model Verification

The powder-feeding gas was considered to be ideal state N_2_, the inlet pressure of the nozzle was 5 MPa, the inlet temperature was set to 380 °C, the powder-feeding gas pressure was set to 5.1 MPa, the powder-feeding gas temperature was set to 25 °C, and the nozzle pressure outside the outlet was set to 5.1 MPa. The environmental pressure and temperature were set to 0.101 MPa and 25 °C, respectively. Finally, the density-based solver was used for the iterative solution process until convergence.

The nozzle cross-sectional temperature and N_2_ velocity distribution programs calculated by the computational fluid dynamics (CFD) software Fluent (ANSYS 2022 R1) are shown in Figure 5a. Figure 5b shows the changes in the velocity and temperature of basalt fiber along the nozzle *X*-axis. Because of its cylindrical structure, basalt fiber is difficult to model with the DPM method, and its unique structure in the Laval nozzle is extremely complex. For this reason, we simplify its model to a spherical DPM model with an equivalent diameter and endow it with relevant physical properties of basalt fiber, such as heat conductivity, specific heat capacity, density, and Young’s modulus. It can be seen that the temperature of the basalt fiber along the *X*-axis reaches a maximum value of 589 K, and the velocity reaches a maximum value of 768 m/s when it is close to the substrate. These data are further used as the particle velocity and temperature in the subsequent basalt fiber deposition simulations. Figure 5c shows the velocity and temperature changes along the axis of the nozzle using N_2_ as the powder-feeding gas. It is clear that the velocity and temperature of N_2_ in the nozzle have a certain oscillation. This can be attributed to the fact that the intrinsic design pressure of the nozzle is 5 MPa, while the expansion ratio is greater than the optimal value, and an oblique shock wave is generated at the exit of the nozzle. However, this oblique shock wave has the strongest influence on the particles with a small diameter (1–5 μm) and has a minor effect on most of the particles used in this experiment with a diameter of 10–100 μm. Figure 5d shows the correlation between the velocity and temperature of the particles (diameter: 10–100 μm) exiting the nozzle outlet and the particle size. In the numerical simulation of fluid–structure interaction using Fluent, the particles are simulated using the discrete phase model (DPM), which does not consider the particle pair reverse effects of airflow.

#### 4.1.1. Al Single-Particle Deposition

To clarify the deformation ability of Al particles impacting the Al substrate, the impact behavior of single particles on the Al substrate was analyzed through the finite element method. In this simulation, the adhesion of particles to the substrate was not considered, and the speed and temperature of particles and the temperature of the substrate surface depended on the relevant CFD simulation results. Figure 6 shows the morphology and simulation results of single Al particles with an average diameter of 55 μm deposited on an Al substrate (substrate temperature: 25 °C) under the parameters of 390 K and 580 m/s. The detailed boundary conditions of particles and substrate are described in Figure 1a.

As shown in Figure 6a, the Al particles and substrate suffer from large plastic deformation. Under the condition of high-speed impact, the particles undergo large deformation and combine with the substrate, and part of the material is squeezed away from the substrate due to the metal jet effect. The numerical simulation results are shown in Figure 6b,c, The maximum local temperature and the maximum local stress of particles reach 728.794 K and 239.367 MPa, respectively. Suresh et al. [40]. examined the microstructural evolution of single-grain Al particles during supersonic shock by quasi-coarse-grained dynamics (QCGD) simulations and identified the critical velocity for thermal softening at the periphery of the particle/substrate interface (i.e., adiabatic shear instability in the substrate/particles). The competing events of stress hardening and local softening at the interface under this velocity were used to explain the temperature and microstructural deformation behavior of Al particles during the deposition process, and it was proved that Al was effectively adaptable to material deformation before the occurrence of metal jets. According to the simulated plastic strain curve in Figure 7b, it can be seen the that the deposition process of Al particles on the substrate involves high plastic strain. When the Al particles have kinetic energy, their plastic deformation rate is significantly higher than that of the substrate material. In the single-particle numerical simulation study by Lin et al. [41]. the influence of interfacial bonding on the impact residual stress of the particles was dominant, which was higher than the impact response of the substrate material. This is consistent with the residual stress distribution shown by single-particle Al powder shock in this study. As shown in Figure 7a, with the passage of the impact time, the stress rapidly rises from 0–10 ns. During this time, the Al particles and the substrate are just in contact, and most of the Al particles still retain their kinetic energy, as shown in Figure 7d. With the progress of the deposition, the stress gradually becomes stable, most of the kinetic energy of the particles is converted into plastic strain energy, and a small part of the kinetic energy is converted into the internal contact elastic energy of Al powder to resist plastic deformation. During the entire deposition process, the particles have a high initial temperature (390 K), most of the heat is transferred to the substrate, and most of the heat generated during the particle impact deformation process is also converted into plastic strain energy. Figure 7c shows the temperature changes in single-particle Al and the bonding part of the matrix with the process of deposition. Due to the large plastic deformation of single-particle aluminum, the ultra-high strain rate leads to a higher temperature change rate than that of the aluminum matrix in the bonding part.

According to the post-processed numerical simulation results for the deposition of single-particle Al powder, the plastic strain rate of Al powder is rather high during the CS deposition process, and the maximum value is $9.7\times 10^{7}\;S^{-1}$, which suggests that the Al powder has a good deformation ability. Schreibe et al. [42] also found that Al particles had a rather high strain rate range, ranging from 10^−3^ to 10^13^, in the impact test of single Al particles, further indicating that Al powder exhibits an excellent deformation performance. In the numerical simulation of the deposition of basalt and Al powder mixed spray, it can be clearly observed that under the condition of high-speed impact, the edge of the particles is squeezed out of the metal jet when the Al powder is in contact with the substrate surface. These extruded Al powders in a high-temperature molten state can be well bonded to other particles and substrates in multi-particle mixed spray, which is beneficial to the adhesion and density of the coating. During the whole process, Al powder perfectly coats the basalt fiber in CS coating through metal interlocking, metal smelting, and other mechanisms and achieves better compactness at the same time, which can effectively improve the deposition efficiency of basalt fiber in the coating.

#### 4.1.2. Cohesive Unit Suitability Verification

To verify the applicability of introducing a cohesive element to the numerical simulation of basalt fiber fracture, we designed the numerical simulation experiments of single-particle aluminum deposition and subsequent single-particle aluminum impact. Cohesive units have been applied to the deposition of cold-sprayed ceramic reinforced phases. Chakrabarty and Song [43] initially used the smoothed particle hydrodynamics (SPH) method to simulate the brittle fracture behavior of ceramic particles after hitting the substrate. However, the ceramic morphology characterized by this method was quite different from the actual experimental results, and the SPH method was found to be mainly suitable for the Euler method. Then, to better compare the simulation and experimental results, Chakrabarty and Song [26] generated grain boundaries in the ceramic particles by the Voronoi seed point method in the subsequent study and inserted condensed units. Additionally, CS ceramic deposits were destroyed.

In the model, we use basalt fiber to deposit parallel to the matrix plane. This is because when basalt fiber is deposited parallel to the matrix plane, its contact area with the matrix surface is the largest, which is the special case most prone to fracture. Of course, the positions and angles of basalt fiber deposition in front of the matrix are various, which we have taken into account in the subsequent numerical simulation of multi-particle mixed deposition.

It can be seen from the post-processing results of the numerical simulation shown in Figure 8b that under the velocities of 500, 600, 700, and 800 m/s, the single basalt fiber does not break after hitting the substrate but rebounds on the substrate surface. Since the substrate is aluminum, which is a relatively soft material, the substrate deformation absorbs most of the energy during the collision of the stronger basalt fiber with the substrate, and the basalt fiber rebounds before reaching the material’s breaking strength of 3773 MPa. Accordingly, the subsequent impact of a single Al particle is examined. The particle diameter is 10–100 μm (based on laser particle size analysis). The average diameter is 55 μm, the particle speed is 600 m/s, and the temperature is 389 K.

According to the variation in the stress of a single basalt particle with time in Figure 8b, the basalt fiber has the largest stress concentration effect when it just touches the substrate. However, most of the energy is absorbed by the substrate after the impact, and the remaining energy of the basalt fiber causes it to rebound on the substrate surface without further increasing the stress to reach the tensile strength of 3773 MPa, so it does not break while the substrate is deforming. It can be seen in Figure 8a that the stress concentration caused by the basalt fiber when it just touches the substrate is equivalent to the stress concentration effect generated under the velocity of 600 m/s in Figure 8b. However, with the further impact of the subsequent Al particles, the stress of the single-particle basalt fiber continues to rise and finally reaches 3891 MPa, which exceeds the tensile strength of the basalt fiber, and the material is then destroyed. Meanwhile, due to the continuous extrusion of Al particles, the basalt fiber does not rebound, and it is covered by Al particles and mechanically locked on the surface in contact with the substrate. To examine the interaction between CS particles, Yin et al. [44] considered the subsequent consecutive impact of multiple particles to find that the effect of subsequent particles on the coating was rather significant. For successively impacted particles, the subsequent incident particles compacted the previously deposited particles, resulting in a coating with pores near the surface and a denser coating inside. This phenomenon verifies the applicability and accuracy of the cohesive parameter of single-particle basalt fiber and also shows that the fracture behavior of basalt fiber in the mixed spray is not only related to its properties, such as speed and temperature, but also to the continuous flow of Al particles. The impact on the basalt fiber morphology is also large.

### 4.2. Al–Basalt Fiber Co-Deposition

In our study, the finite element numerical analysis method of Fluent gas dynamics combined with ABAQUS explicit dynamics used has some limitations. The basalt fibers leaving the Laval nozzle have velocities in each degree of freedom direction, and therefore their morphologies are diverse. Modeling the analysis according to its velocity and morphology in each degree of freedom is a problem with a very large and difficult workload. We simplify the morphological diversity of basalt fibers by creating multiple poses of basalt fibers. Additionally, since the normal component of the velocity of the basalt fiber is the most dominant, we replace its velocity load with the normal component of the velocity.

For the simulation of multi-particle random deposition, the velocity and temperature of Al particles and basalt fiber before deposition are based on the CFD simulation data in the previous section. Next, a 15 vol.% and 30 vol.% basalt fiber-Al powder multi-particle random deposition simulation model has been established. According to laser particle size analysis, the number of Al particles and basalt fibers in the finite element model is shown in Figure 9a,b.

Herein, the average diameter of basalt fiber was 15 μm, and the lengths were in the range of 60–148 μm (60, 80, and 100 μm), which were measured through the SEM images, and the kinetic analysis step was 650 ns.

To better observe the distribution of basalt fibers in the coating and avoid the inevitable loss of the fibers during the sample preparation process, the coated section is exposed by breaking, The SEM images and EDS analysis of the composite coating cross-section and surface are shown in Figure 10.

According to the simulated deposition morphology (Figure 11a,b), obvious metal smelting and mechanical interlocking effects exist between the Al particles (Al, basalt fiber deposition process dynamic video, can be downloaded from the support material link.). Through the SEM images of the fracture surface of the coating, we found the basalt fiber deposition morphology corresponding to that in the numerical simulation results, as shown in Figure 11c–f. In the process of Al particle deposition, due to the adiabatic shear instability of the Al material, the plastic strain energy is dissipated, leading to the local temperature rise of the contact part between the Al particles and the matrix, inducing the thermal softening of the Al material in the contact part between the particles and the matrix, resulting in a jet effect similar to the flow of viscous materials.

Due to the jet effect of the Al material in contact with the substrate, the thermally softened Al particle materials contact each other. Some basalt fibers dispersed in aluminum powder materials bounce off the matrix after contacting the matrix, and some are covered by aluminum materials that are thermally softened and combined with each other. The particle materials without the metal jet effect will form a relatively closed space. A part of basalt fiber is locked in it. At the same time, due to the continuous impact of aluminum powder materials, the kinetic energy of particle impact causes basalt fiber fracture.

As shown in Figure 12a, shows the stress changes in three basalt fibers of different lengths that first contact the matrix during the whole deposition process. Due to the impact of aluminum powder particles, the stress of basalt fibers oscillates. In the process of stress oscillation, fatigue damage caused by stress accumulates continuously. When the stress exceeds the breaking strength of the basalt fiber, the basalt fiber breaks, and the stress oscillation tends to decrease, which may be related to the change in the position of the basalt fiber under the impact of the aluminum particle kinetic energy. At the same time, the basalt fiber with a length of 100 μm reached the breaking strength at the earliest time, approximately 200 ns. The time for basalt fiber of 80 μm in length to reach the breaking strength is next, approximately 280 ns, and the time for basalt fiber of 60 μm in length to reach the breaking strength is the latest, approximately 320 ns.

Figure 12c shows the strain evolution of three basalt fibers of different lengths during the deposition process. It can be seen that with the progress of the deposition process, the strain value will have a sudden change, and the time of the sudden change coincides with the time when the stress reaches the fracture strength as shown in Figure 12a. These phenomena indicate that the length of basalt fiber will affect its existence form after deposition, and shorter basalt fiber is easier to maintain its pre-deposition structure after deposition. Figure 12c,d shows the stress and strain changes in the substrate in contact with basalt fibers of three selected lengths during deposition and the temperature changes in the parts in contact with the substrate. Due to the impact of the deposited aluminum powder and basalt fiber, the kinetic energy of particles and fibers is absorbed by the matrix, resulting in the deformation of the matrix, the temperature is increased, and the stress value of the matrix is also subject to vibration during the deposition process.

Similarly, for mixed spraying with a basalt fiber content of 30 vol.%, we also selected three lengths of basalt fiber that first contacted the substrate. As shown in Figure 13a,c, the stress and strain changes in the three lengths of basalt fibers during the deposition process are shown. Compared with mixed spraying of 15 vol.% basalt fiber, the spraying basalt fiber content of 30 vol.% reaches the breaking strength earlier, and the stress value is more violent. Due to the increase in basalt fiber content, it is inevitable that basalt fibers will contact each other during the deposition process. The contacted basalt fibers will reach the fracture strength faster under the continuous impact of aluminum powder particles.

After the post-processing of the numerical simulation results, the basalt fibers in the coating show four different morphologies: fracture, transverse crack, bent, and deformed to a small extent. Due to the thermal softening of local materials induced by the adiabatic shear instability effect, this part of aluminum materials produces jet phenomenon and combines during the bonding process with the matrix. Due to the thermal softening of local materials induced by adiabatic shear instability effect, during the bonding process between aluminum particles and the matrix, the metal aluminum with local thermal softening will produce a metal jet effect similar to the viscous material flow phenomenon, and the flowing aluminum metals will bond with each other.

At the same time, there are two situations: first, the thermal softened aluminum metal will cover the basalt fiber; second, the aluminum metal of the non-softened part will form a closed space to lock the basalt fiber. Under the continuous impact of subsequent aluminum particles, due to the low hardness of the thermally softened aluminum metal, the basalt fiber coated with aluminum will directly receive most of the impact kinetic energy of the particles, causing the basalt fiber with transverse cracks to continue to deform, and then a brittle fracture occurs. The aluminum metal forming the enclosed space has a certain strength, which can offset part of the kinetic energy impact. Therefore, basalt fibers locked by the confined space can maintain the existence of transverse cracks, bending, and other morphologies.

The basalt fibers extracted from the numerical simulation results are shown in Figure 14. We found that basalt fibers characterized by bending and transverse cracks only appear in the coating area forming an enclosed space. Among the curved basalt fibers, most of them are 100 μm long, and only a few of them are 60 and 80 μm long. The basalt fiber covered by the heat-softened aluminum metal has the characteristics of brittle fracture, and most of the 80 and 100 μm basalt fibers split into three segments after brittle fracture, while the 60 μm basalt fibers generally split into two segments. This shows that the aspect ratio of basalt fiber will affect its morphology after deposition. Similarly, being in different positions of the coating (such as material covering and confined space locking) also affects the morphology of basalt fiber.

To explore the influence of the long-diameter ratio on basalt fiber deposition, a mixed-deposition experiment with a diameter of 12 μm and six long-diameter ratio gradients (2:1, 3:1, 4:1, 5:1, and 6:1) was designed. A total of 100 basaltic fiber mixed depositions of 10–100 μm diameter Al particles were employed, and the relevant powder particle temperature and velocity load were determined according to previous CFD numerical simulations.

As shown in Figure 15, based on numerical simulations, we found that there was no brittle fracture of basalt fibers with a length-to-diameter ratio of 2:1. Cracks occur when the aspect ratio is 3:1, and when the length ratio is greater than 3:1, the basalt fibers are brittly fractured. The results show that the short-cut basalt fiber can reduce and avoid cracks and brittle fractures by adjusting the aspect ratio. Related scholars have found that common ceramic-reinforced phases such as Al_2_O_3_ [16], WC-25Co [45], WC-17Co [22], and SiC [17] show brittle fracture and cracking patterns after deposition in composite coatings. This is similar to the brittle fracture and transverse cracking that occurs when basalt fibers are deposited. However, the particle morphology determines that bending and deformation do not occur, which is a deposition morphology unique to basalt fiber deposition influenced by the aspect ratio.

From the numerical simulation results after the deposition of basalt fiber with two contents in Figure 16, we found that increasing the basalt fiber content in the mixed raw materials can indeed increase its probability of deposition. However, the two deposition forms of material covering and confined space locking also indicate that the uniform dispersion of basalt fibers in mixed raw materials also plays an important role in improving the deposition probability of basalt fibers.

According to the simulated 1/2X cross-sectional strain and stress cloud (Figure 16), the basalt fibers deposited in the coating are completely covered by molten Al powder and are mechanically locked, which is consistent with Figure 14. This SEM images of related samples in Figure 16b,d are in agreement with this observation. According to the simulated cross-sectional equivalent plastic strain of coating with 15 vol.% basalt fiber (Figure 16a), the deposited basalt fibers have fewer voids around the coating, and the coating is relatively dense. However, under the equivalent plastic strain of the coating section with 30 vol.% of basalt fiber (Figure 16c), more basalt fiber deposition is observed, but the pores around the basalt fibers are larger, and the deformation degree of Al powder is higher than that in the bigger coating with 15 vol.% basalt fiber. This indicates that the increase in the content of basalt fiber in the mixed powder affects the density and energy of mixed powder deposition.

### 4.3. Coating Performance

The hardness of the coatings with basalt contents of 15 vol.% and 30 vol.% was measured by a Rockwell hardness gauge (Model: FangYuan XHRD-150). To ensure the accuracy of the results, 10 test points were randomly selected on the coating surface with a size of 40 × 30 mm^2^. Data were collected for each test point, and the average Rockwell hardness of the two coatings was obtained.
(10)$$HRC=N-\frac{\Delta h}{s}$$
where Δh is the residual indentation depth increment under the initial testing force F0 after removing the main testing force; the N value of the ball’s Rockwell scale is 130; C is constant, generally C = 0.002 mm. The relevant experimental data are shown in Figure 17.

According to the data of the Rockwell hardness test, the Al–basalt powder composite coating has a high hardness, and with the increase in the basalt content, the Rockwell hardness increases from HRC38.9 to HRC49 Its hardness is similar to that of WC-17Co composite coating prepared by Emmanuel et al. [22]. This suggests that the basalt fiber reinforcement phase improves the hardness of the composite coating.

In the friction and wear test data, the stable value of the dynamic friction coefficient is close to the WC-25Co wear-resistant coating prepared by Dosta et al. [45]. Figure 18a,b show the friction-wear test results, which reveal an interesting phenomenon. Specifically, as the load increases, the kinetic coefficient of friction of the composite coating shows a decreasing trend, which is the opposite of the behavior in the general metal friction-wear test. Similarly, under the same load condition, the kinetic coefficient of friction of composite coating with 30 vol.% basalt fiber is higher than that of coating with 15 vol.% basalt fiber.

We found that the dynamic friction factor of the composite coating in the wear test had oscillations, especially in the composite coating with 30% high basalt fiber content. In addition, with the increase in the load applied by the spherical grinding head in the wear experiment, the vibration tends to decrease and gradually stabilize. Therefore, we observed and characterized the worn surface of the composite coating by SEM, and the results are shown in Figure 19.

As shown in Figure 19a, the color of the coating compacted by the spherical grinding head in the wear experiment is white. With the reciprocating motion of the spherical grinding head, the coating starts to wear gradually. In this process, the enclosed space formed by no thermally softened Al particles is gradually exposed in the form of pits. At the same time, the basalt fibers in the coating also begin to be exposed and fall off, leaving pits at the falling position, as shown in Figure 19b. This will change the surface roughness of the coating and make the dynamic friction coefficient vibrate. Different from the metal material wear test, this vibration will exist until the coating is worn through and contacts the metal substrate. At the same time, the detached basalt is in constant contact with the spherical grinding head in the reciprocating motion. Repeated rolling reduces the size of basalt fiber, similar to the wear particles in the wear experiment. Some studies have shown that when the size of abrasive particles is less than 50 μm [46], the sliding friction will change into rolling friction, which will reduce the dynamic friction coefficient. This can also explain the problem that the dynamic friction coefficient decreases before the coating wears through. With the increase in basalt content, the number of basalt shed and exposed in the wear test will also increase, resulting in more pits on the coating surface, making the vibration of the dynamic friction coefficient more obvious.

## 5. Conclusions

In this study, the deposition behavior of Al–basalt fiber blends on Al substrates was systematically investigated by a combination of numerical simulations and experiments based on CFD and the CEL method. We found that during the process of forming a composite coating by mixing aluminum and short-chopped basalt fibers, aluminum, and basalt fibers have two main modes of interaction. First, aluminum that undergoes thermal softening produces a metal jet effect, which encapsulates the basalt fibers. Second, aluminum that does not undergo thermal softening forms a closed space with each other, trapping the basalt fibers inside. Meanwhile, basalt exists in the composite coating in four forms—bending, deformation, transverse cracks, and brittle fracture—under these two modes of contact with aluminum. In addition, the Rockwell hardness tests and the friction-wear tests show that adding basalt fibers can improve the hardness and wear resistance of the composite coating.

## Figures and Tables

**Figure 1 materials-16-01862-f001:**
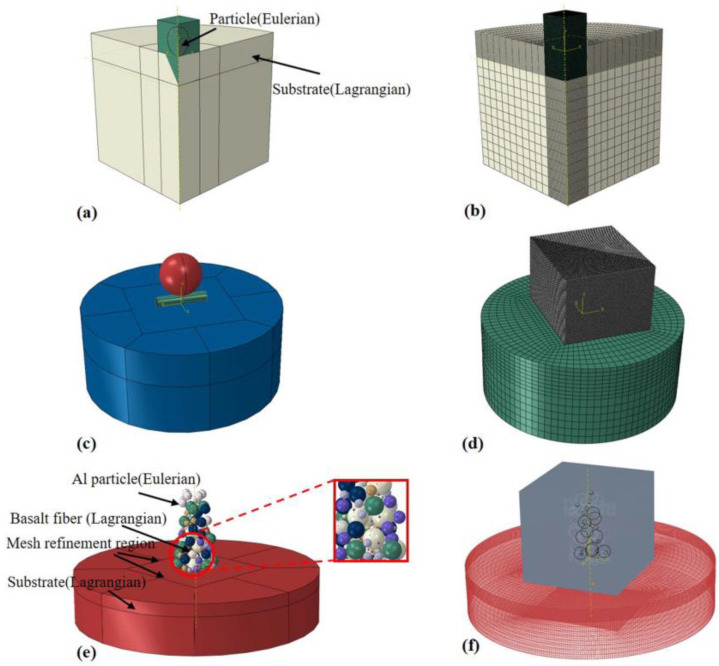
(**a**,**b**) Al single-particle CEL model and its meshing; (**c**,**d**) single-particle basalt fiber and single-particle Al impact CEL model and meshing; (**e**,**f**) random impact model of Al–basalt fiber mixed with multiple particles and meshing.

**Figure 2 materials-16-01862-f002:**
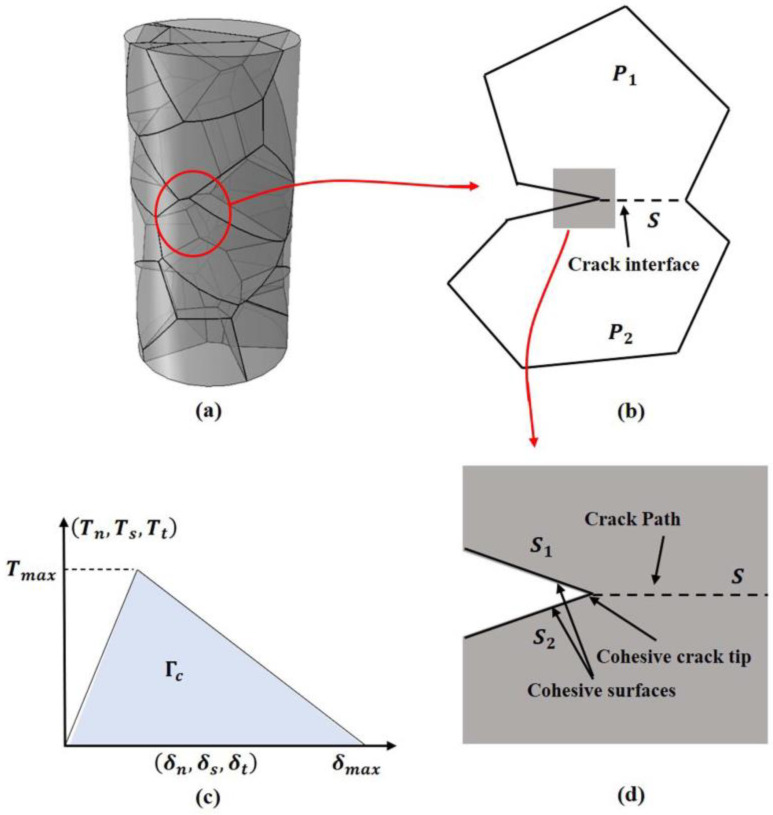
(**a**) Basalt model based on Voronoi random seeds. (**b**) crack propagation along the polyhedral surface. P1 and P2 are two different cross sections of the polyhedra in contact with each other, and s is the initial mutual contact surface. (**c**) Traction separation laws for cracks and damage of the material, where $\left(T_{n},T_{s},T_{t}\right)$ and $\left(\delta_{n},\delta_{s},\delta_{t}\right)$ are the common traction and separation components, and $T_{max}$ and $\delta_{max}$ represent the maximum traction force due to damage and the maximum separation displacement at the damage, respectively. (**d**) Local amplification of crack propagation.

**Figure 3 materials-16-01862-f003:**
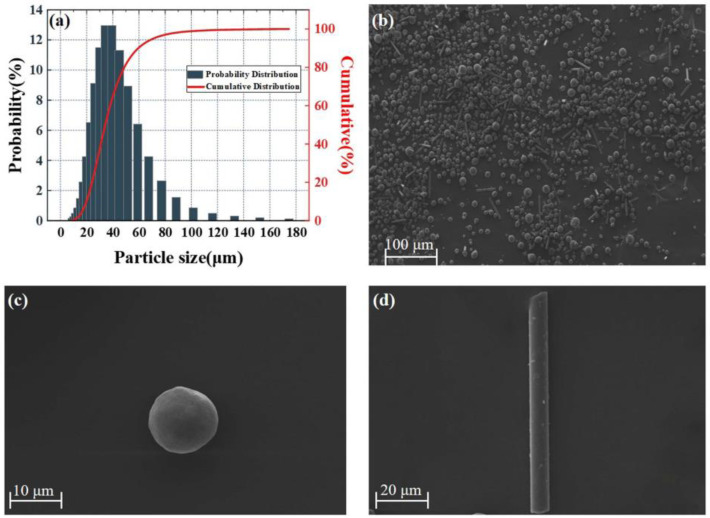
(**a**) Particle size distribution of aluminum powder; (**b**) SEM image of basalt–pure aluminum mixed powder; (**c**) single-particle Al particle; (**d**) single-particle basalt fiber.

**Figure 4 materials-16-01862-f004:**
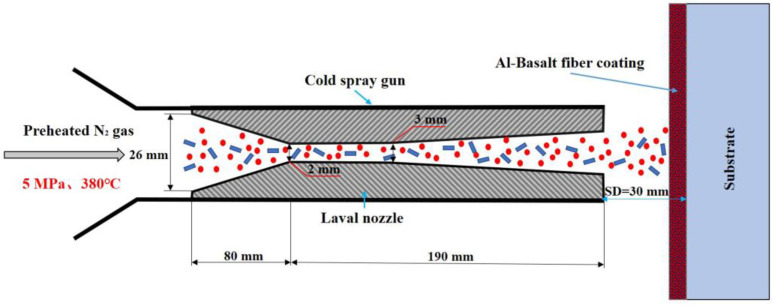
Cold spray process of Al–basalt fiber hybrid.

**Figure 5 materials-16-01862-f005:**
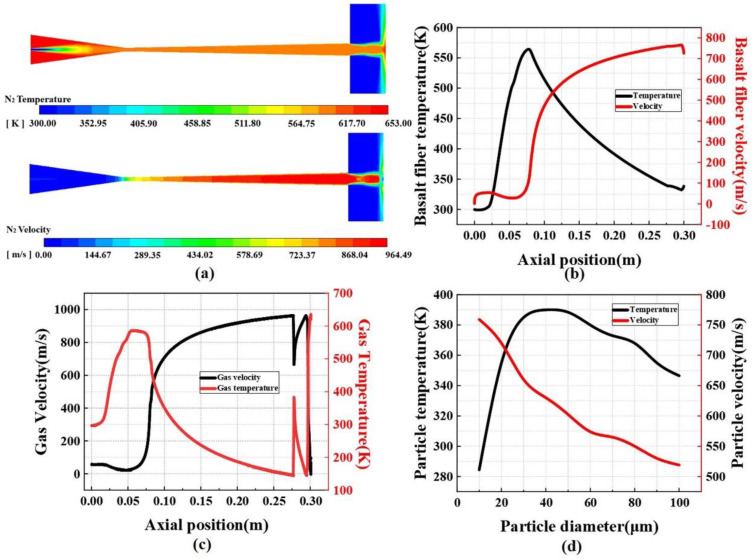
(**a**) Temperature and velocity variation of N2 in the nozzle. (**b**) Variation of CBF velocity and temperature along the *X*-axis. (**c**) Variation of velocity and temperature of N2 along the *X*-axis. (**d**) Velocity and temperature of aluminum powder with different diameters.

**Figure 6 materials-16-01862-f006:**
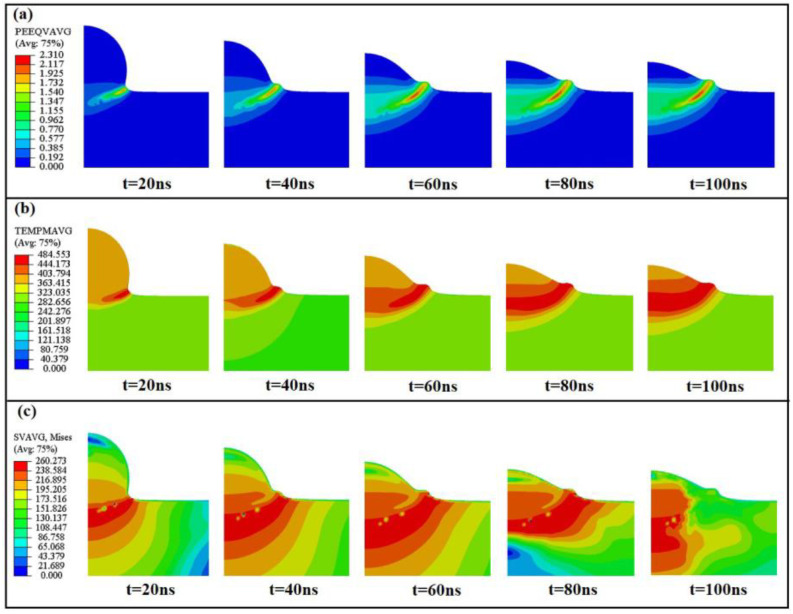
The simulation results show (**a**) the evolution of the equivalent plastic strain, (**b**) the evolution of the temperature and (**c**) the evolution of the stress.

**Figure 7 materials-16-01862-f007:**
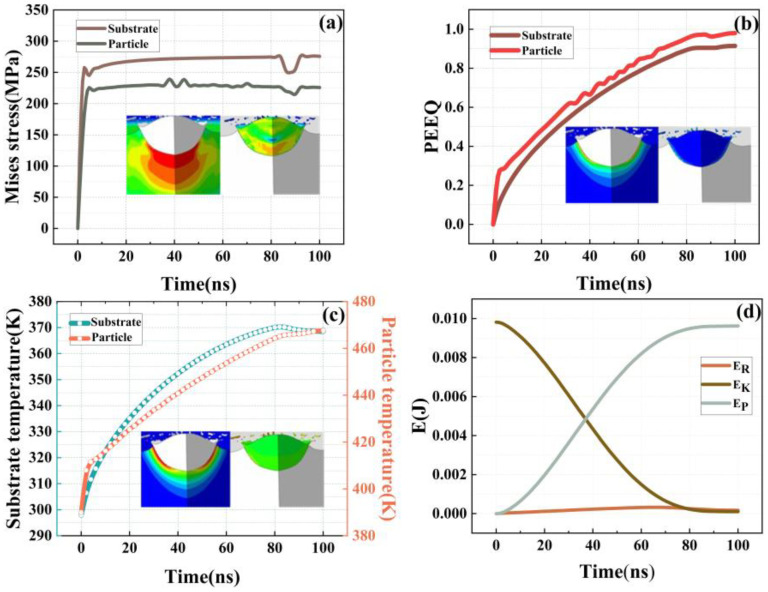
Post-processed simulation results. (**a**) Stress evolution of aluminum particles and matrix with time. (**b**) Evolution of plastic strain of aluminum particles and matrix with time. (**c**) Evolution of temperature of aluminum particles and matrix with time. (**d**) Evolution of overall kinetic energy (EK), plastic strain energy (EP), and internal contact elastic energy (ER) with time.

**Figure 8 materials-16-01862-f008:**
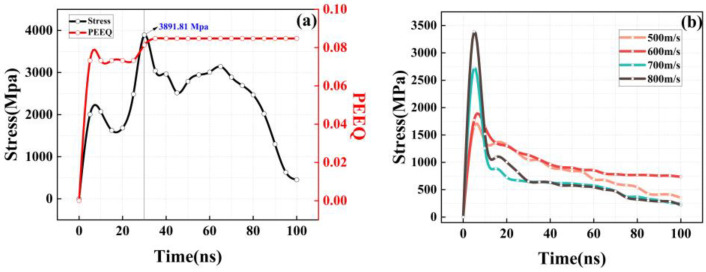
Impact simulation of single-particle basalt fiber; (**a**) single-particle impact of Al–basalt fiber; (**b**) single-particle impact of basalt fiber under the velocities of 500, 600, 700, and 800 m/s.

**Figure 9 materials-16-01862-f009:**
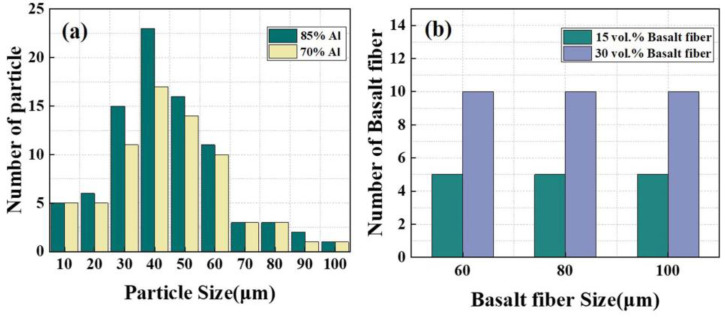
Al–basalt fiber multi-particle random deposition simulation. (**a**) Particle size distribution of Al powder. (**b**) Number and length distribution of basalt fiber.

**Figure 10 materials-16-01862-f010:**
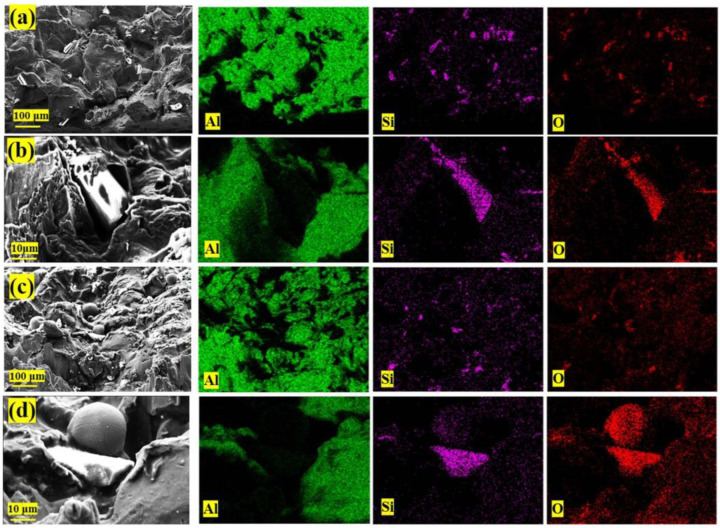
SEM and EDS analysis images of the composite coating. (**a**) Coated cross-section; (**b**) closed space locked CBF; (**c**) coated surface; (**d**) CBF clad with thermos-softened Al.

**Figure 11 materials-16-01862-f011:**
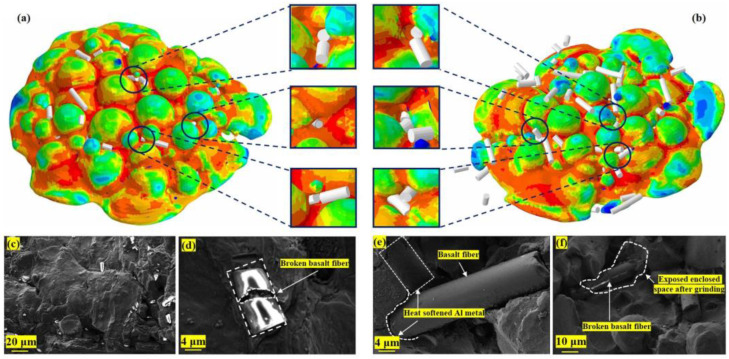
Stochastic deposition simulation of Al–basalt fiber mixed multi-particle. (**a**) Deposition with 15 vol.% CBF; (**b**) CBF 30 vol.% deposition; (**c**) composite coating cross-section; (**d**) fractured CBF; (**e**) trace pit after CBF exfoliation; (**f**) CBF with cracks.

**Figure 12 materials-16-01862-f012:**
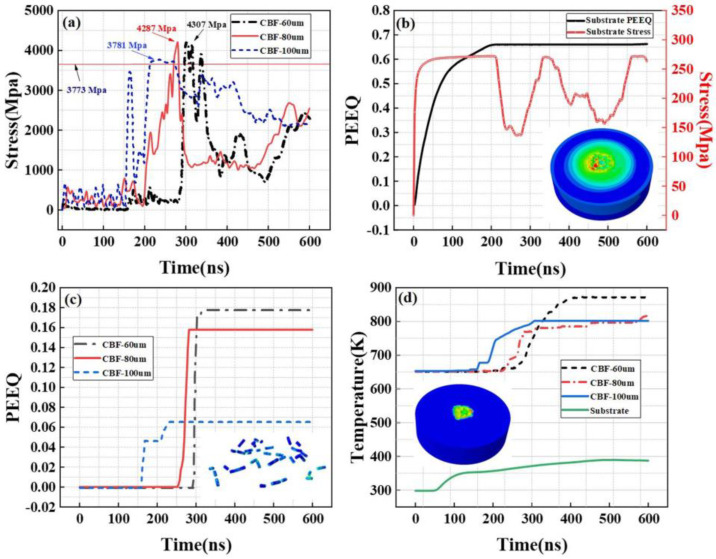
Simulation of the random deposition of Al-basalt fibers in multi-particles for a CBF content of 15 vol.%. (**a**) CBF stress evolution; (**b**) evolution of the surface stress and equivalent plastic strain in the aluminum matrix; (**c**) CBF equivalent plastic strain evolution; (**d**) temperature evolution of the aluminum matrix and basalt fibers.

**Figure 13 materials-16-01862-f013:**
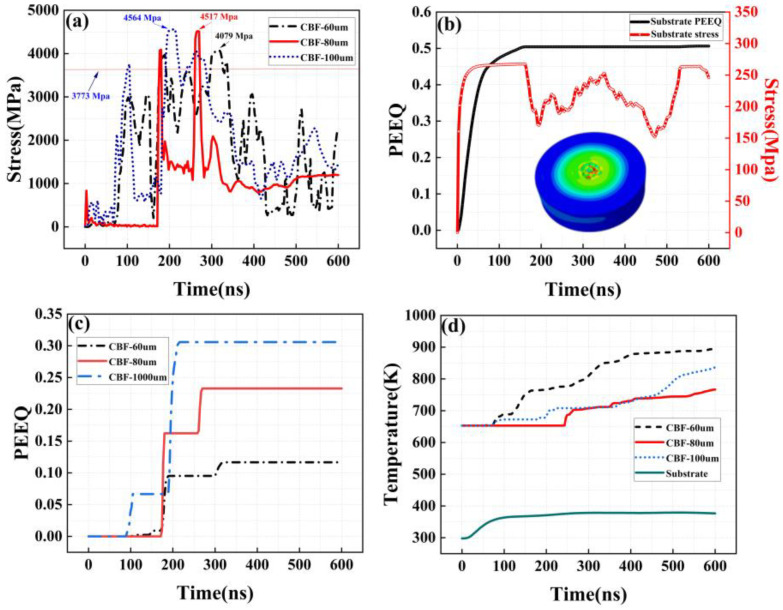
Simulation of the random deposition of Al–basalt fibers in multi-particles at a CBF content of 30 vol.%. (**a**) CBF stress evolution. (**b**) Evolution of the surface stress and equivalent plastic strain in the aluminum matrix. (**c**) CBF equivalent plastic strain evolution. (**d**) Temperature evolution of the aluminum matrix and basalt fibers.

**Figure 14 materials-16-01862-f014:**
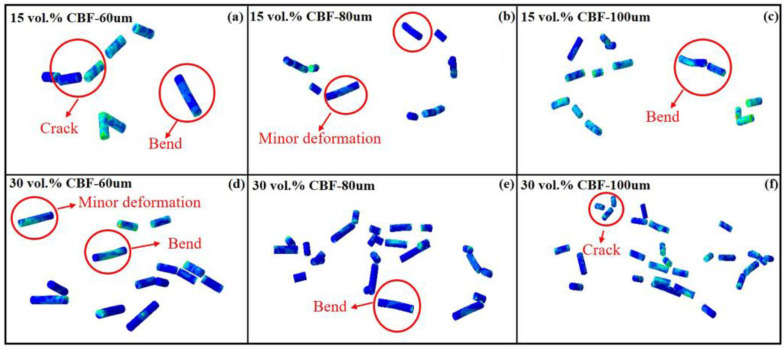
Basalt fibers in the mixed multi-particle simulated coating. The 15 vol.% CBF composite coating: (**a**) 60 μm; (**b**) 80 μm; (**c**) 100 μm. The 30 vol.% CBF composite coating: (**d**) 60 μm; (**e**) 80 μm; (**f**) 100 μm.

**Figure 15 materials-16-01862-f015:**
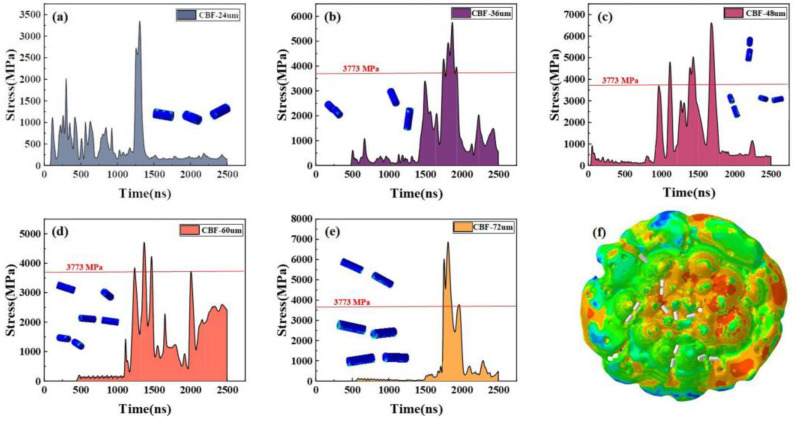
Basalt fiber stress over time: (**a**) 24 μm CBF; (**b**) 36 μm CBF; (**c**) 48 μm CBF; (**d**) 60 μm CBF; (**e**) 72 μm CBF; (**f**) composite coating bottom.

**Figure 16 materials-16-01862-f016:**
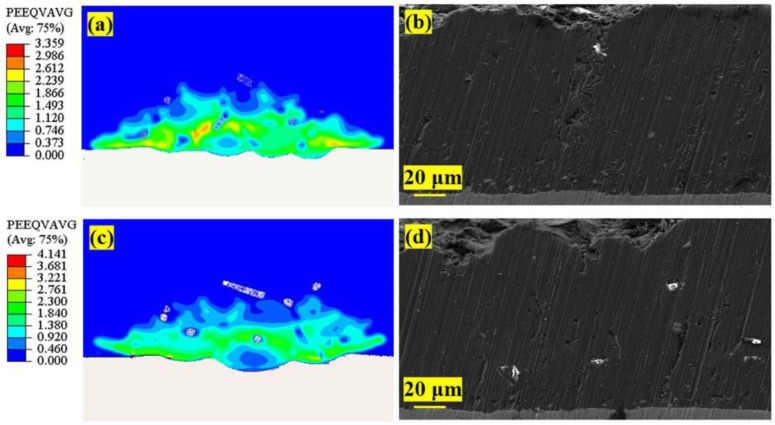
Composite coating cross-section: (**a**) 15 vol.% CBF simulation cross-section; (**b**) SEM picture of 15 vol.% CBF cross-section; (**c**) 30 vol.% CBF simulation cross-section; (**d**) SEM picture of 30 vol.% CBF cross-section.

**Figure 17 materials-16-01862-f017:**
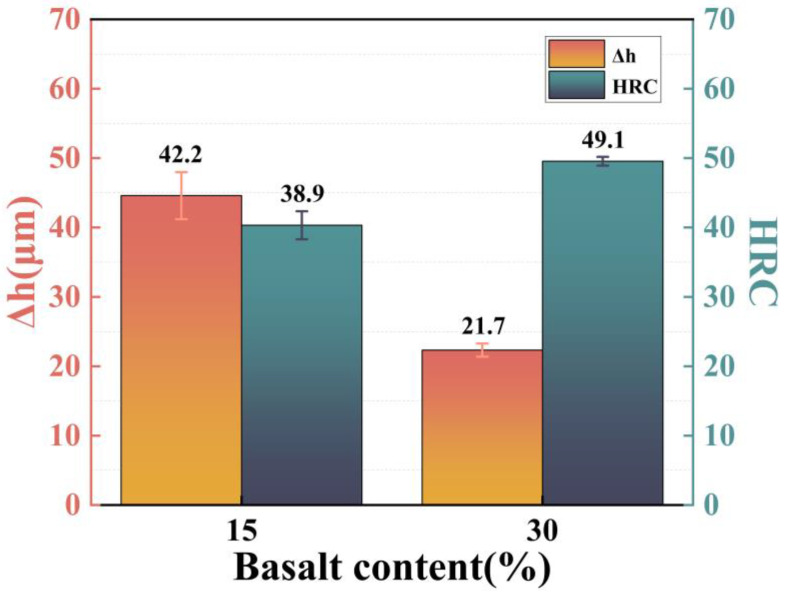
Rockwell hardness test results of Al–basalt powder composite coating.

**Figure 18 materials-16-01862-f018:**
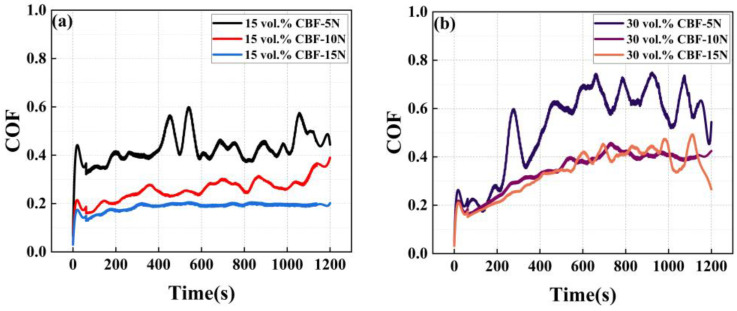
Friction coefficient test results of 15 vol.% and 30 vol.% basalt fiber composite coatings: (**a**) evolution of dynamic friction coefficient of 15 vol.% basalt content composite coating under different loads; (**b**) evolution of the coefficient of kinetic friction of 30 vol.% basalt content composite coating under different loads.

**Figure 19 materials-16-01862-f019:**
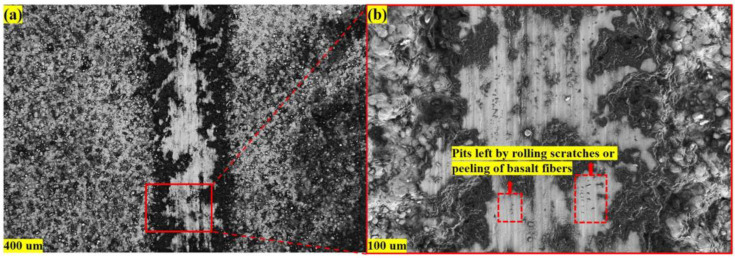
(**a**) SEM image of the wear scratch of the composite coating; (**b**) partial magnification of the scratch.

**Table 1 materials-16-01862-t001:** Material parameters.

	Parameter (Unit)	Basalt Fiber	Al
General	Density, $\rho \;\left(\mathrm{kg}/m^{3}\right)$	2700	2710
	Specific heat, $C_{P}\;\left(J/\mathrm{kg}\cdot K\right)$	940	910
	Thermal conductivity (W/m·K)	0.013	237
	Thermal expansion $\left(10^{-6}/K\right)$	140	16.4
	Melting temperature, $T_{m}\left(K\right)$	1473	916
	Inelastic heat fraction, β	0.9	0.9
Elastic	Elastic modulus, GPa	94	65.8
	Poisson’s ratio	0.3	0.3
Plastic (JC plasticity model)	Yield stress, A (MPa)	—	148.4
	Hardening constant, B (MPa)	—	345.5
	Hardening exponent, n	—	0.183
	Strain rate constant, C	—	0.001
	Thermal softening exponent, m	—	0.895
	Modified JC model material constant, λ	—	—
	Reference strain rate, ${\dot{\varepsilon }}_{0}\left(1/S\right)$	—	1
	Reference temperature, $T_{ref}\left(K\right)$	—	298

**Table 2 materials-16-01862-t002:** Cohesive data of basalt fiber.

Physical Quantity.	Numerical Value	Unit
Rigidity	3.13 × 10^9^	MPa/mm
Critical strain	1.08 × 10^−6^	mm
Fracture strain	5.4 × 10^−5^	mm
Fracture energy	0.09118	J
Critical thickness of the fracture	3 × 10^−5^	mm

**Table 3 materials-16-01862-t003:** Deposition parameters and characteristics of the CGDS process.

Cold Spray PCS1000	
Powder injection position	Powder concentration bin (0.5 L)
Nozzle material	Alloy steel
Maximum temperature	1000 °C
Maximum pressure	5 MPa
Spraying distance (SOD)	30 mm

**Table 4 materials-16-01862-t004:** Experimental friction and wear parameters.

Movement	Reciprocating Motion		
Line speed	10 mm/min		
Load	5 N	10 N	15 N
Load time	20 min		
Frequency	2 Hz		
Grinding head diameter	5 mm		
Grinding head material	440C		

## Data Availability

The relevant research data has been presented in this article, in addition to this, authors can only provide shared data with Appendix A.

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
