# Peer review of "Numerical and Experimental Investigations of Cold-Sprayed Basalt Fiber-Reinforced Metal Matrix Composite Coating"

_materials, 2023, doi:10.3390/ma16051862_

Round 1

Reviewer 1 Report

Presented paper is very interesting, and can be published after minor revision.

What source of Al and basalt fibres?

Informations about SEM, EDX, cross-section analysis should be added in a methodology part.

Author Response

Cover letter-responses to reviewer comments

Dear Editors and Reviewers:

Thank you so much for your letter and for the reviewers’ comments and suggestions to our manuscript entitled “Numerical and Experimental Investigations of Cold Sprayed Basalt Fiber Reinforced Metal Matrix Composite Coating”. These comments and suggestions have been invaluable in helping us to improve the quality of our work. We have meticulously reviewed the manuscript based on your feedback and have taken the necessary steps to address all the issues you have raised. Our responses to your comments are as follows:

Reviewer 1

1.Presented paper is very interesting, and can be published after minor revision

Response: Thank you for your positive evaluation of our paper.

2.What source of Al and basalt fibers?

Response: In this study, pure aluminum powder (New Material Co, Ltd, Guangdong) was used. Basalt fibers were provided by Aerospace Tuoda Basalt Development Co., Ltd (Dazhou, Sichuan).

The information has been added to Line 283 in revised manuscript.

3.Information about SEM, EDX, cross-section analysis should be added in a methodology part.

Response: Thanks so much for your suggestion. In Chapter III "Experimental" and Section 4.3 "Coating performance", the location of relevant instruments and equipment appears for the first time, We have added the information of SEM, EDS, Laser particle size analyzer, Rockwell hardness gauge, and Friction-wear tester. SEM (German ZEISS GeminiSEM 300), EDS (OXFORD Xplore). Laser diffraction analysis (Better BT-9300SE). Rockwell hardness gauge (FangYuan XHRD-150). Friction-wear tester (Bruker UMT TriboLab)

The sentences have been added to Line 281, Line 288, Line 320, and Line 651 in revised manuscript.

We extend our sincere thanks to you for your diligent review of our work. Your constructive feedback and recommendations have been extremely valuable in helping us improve our manuscript. Your insights have guided us in making substantial revisions that have enhanced the quality of our research. We are grateful for the opportunity to benefit from your expertise and are eager to resubmit the revised manuscript for your further review. Once again, thank you for your support and dedication to our work. 

Sincerely,

Yingying Wang

Reviewer 2 Report

The present work entitled " Numerical and Experimental Investigations of Cold Sprayed Basalt Fiber Reinforced Metal Matrix Composite Coating " considers obtaining of the aluminum-basalt fiber composite coating using cold spraying technology.

The article is well written and of great practical interest.

Below are some comments on the article.

1. It is difficult to understand what these modes of interaction between aluminum and basalt mean: “aluminum metal thermal softening cladding and closed space locking”. (line 27)

2. What does "good" hardness and wear resistance mean? (line 28)

3. “(Reviewer: 1 Q2)” (line 235)

4. What counterbody was used in the friction-wear test?

5. In Figure 5, a, it is hard to see the distribution of which parameters the model demonstrates.

6. It was established from the calculations that the aluminum temperature does not exceed 728 K (line 324) or 900 K (Fig. 13,d), while the conclusions indicate local melting of the aluminum powder (line 647). Aluminum melting point is 933 K.

The article is recommended for publication after minor corrections.

Author Response

Cover letter-responses to reviewer comments

Dear Editors and Reviewers:

Thank you so much for your letter and for the reviewers’ comments and suggestions to our manuscript entitled “Numerical and Experimental Investigations of Cold Sprayed Basalt Fiber Reinforced Metal Matrix Composite Coating”. These comments and suggestions have been invaluable in helping us to improve the quality of our work. We have meticulously reviewed the manuscript based on your feedback and have taken the necessary steps to address all the issues you have raised. Our responses to your comments are as follows:

Reviewer 2

The present work entitled " Numerical and Experimental Investigations of Cold Sprayed Basalt Fiber Reinforced Metal Matrix Composite Coating " considers obtaining of the aluminum-basalt fiber composite coating using cold spraying technology. The article is well written and of great practical interest. Below are some comments on the article:

1. It is difficult to understand what these modes of interaction between aluminum and basalt mean: “aluminum metal thermal softening cladding and closed space locking”. (line 27)Response: Thank you for your valuable comments. We apologize for the confusion caused by the terminology used in the paper. We have carefully revised the description of contact modes between aluminum and basalt fibers. We believe that the revised description more accurately reflects the two contact modes and provides a clearer understanding of the deposition process of the Al-basalt fiber composite coating:

Original: Line 31. There are two modes of interaction between basalt fibers and aluminum: aluminum metal thermal softening cladding and closed space locking.

Correction: Line 28. At the same time, there are two modes of contact between aluminum and basalt fibers. Firstly, the thermally softened aluminum encapsulates the basalt fibers, forming a seamless connection. Secondly, aluminum particles that have not undergone softening effect create a closed space, trapping the basalt fibers inside.

2. What does "good" hardness and wear resistance mean? (line 28)

Response: Thank you for your comment. We apologize for any confusion caused by the use of the term "good" when referring to the hardness and wear resistance of the composite coatings. To clarify, in the context of our study, "good" hardness and wear resistance refer to composite coatings that have a higher hardness and wear resistance compared to the benchmark materials used in our experiments.Original: Line 28. The composite coating has good wear resistance and hardness.Correction: Line 37. Moreover, Rockwell hardness test and friction wear test were conducted on Al-basalt fiber composite coating, and the results showed that the composite coating has high wear resistance and high hardness.

3. “(Reviewer: 1 Q2)”

Response: Thank you for your feedback. We apologize for the inclusion of meaningless notes in the line. We understand the importance of clarity and conciseness in scientific writing and we will make sure to carefully review and remove any such notes in the revised version of our paper.

Thank you for bringing this to our attention.

4. What counter body was used in the friction-wear test?

Response: Thank you for your valuable feedback and questions. In regards to the friction-wear test, we would like to clarify that the counter body used was a 440C stainless steel ball with a diameter of 5 mm. This was chosen as the counter body due to its widely used and recognized as a common material for friction-wear testing. We apologize for any confusion and hope this clarification helps to better understand the test results. The relevant additional information we add in Table 4 (Line 322):

Table 4. Experimental friction and wear parameters

Movement

Reciprocating motion

Line speed

10 mm/min

Load

5 N

10 N

15 N

Load time

20 min

Frequency

2 Hz

Grinding head diameter

5 mm

Grinding head material

440C

5. In Figure 5, a, it is hard to see the distribution of which parameters the model demonstrates.

Response: Thank you for your feedback and observations on Figure 5a. We apologize for the difficulty in understanding the distribution of parameters. We have revised the image as follows (Line 339):

Figure 5. (a) Temperature and velocity variation of N2 in the nozzle ; (b) Variation of CBF velocity and temperature along the X-axis ; (c) Variation of velocity, and temperature of N2 along X-axis ; (d) Velocity, and temperature of aluminum powder with different diameters.

6. It was established from the calculations that the aluminum temperature does not exceed 728 K (line 324) or 900 K (Fig. 13,d), while the conclusions indicate local melting of the aluminum powder (line 647). Aluminum melting point is 933 K.

Response: Thank you for your valuable comments. We apologize for the inaccuracy of the description in this part of the conclusion, where "melting" should be corrected to "thermal softening". It is well known that the advantage of cold spray technology is that the coating is formed at temperatures above room temperature and below the melting point of the material. Under high-speed impact, the part of the material in contact with the substrate will have a thermal softening effect similar to material flow due to adiabatic shear instability, but the temperature will not exceed the material melting point temperature. Correction of incorrect wording in the original article is as follows.

Original: Line 713. The local melting of aluminum powder coats the basalt fibers due to the jet effect, but the hardness of the molten metal is low.

Correction: Line 694. First, aluminum that undergoes thermal softening produces a metal jet effect, which encapsulates the basalt fibers.

We extend our sincere thanks to you for your diligent review of our work. Your constructive feedback and recommendations have been extremely valuable in helping us improve our manuscript. Your insights have guided us in making substantial revisions that have enhanced the quality of our research. We are grateful for the opportunity to benefit from your expertise and are eager to resubmit the revised manuscript for your further review. Once again, thank you for your support and dedication to our work.

 Sincerely,

Yingying Wang

Reviewer 3 Report

This paper talks about Numerical and Experimental Investigations of Cold Sprayed Basalt Fiber Reinforced Metal Matrix Composite Coating. The following points need to be clarified:

It is suggested that the authors should modify Section 1 to clearly mention the goal and novelty of this work (mentioned but should be more explicit) and it is important to place the major hypothesis.

The definition of UMT in the abstract should be given.

From which refs. were the values in Table 1 taken?

There should be space before the referencing number. “change[2–5]” should read “change [2–5]”. Implement this throughout the paper.

Also “100um” should read “100 um”. Implement this throughout the paper

The paper lacks a discussion on the weakness and limitations of the present methodology/study.

How was the numerical model verified with the experiments?

The conclusion was too long. It needs to be shortened just to highlight the main findings.

The findings in Section 4 should be supported/compared by similar studies in the literature.

Avoid repeatition in Section 2.1. See below

“In this study, a similar approach has been adopted using the Lagrangian method to model the particles and inserting zero-thickness cohesive elements into the 3D model of the basalt fiber.”

“A zero- thickness cohesive element (Cohesive0) was inserted into the finite element model of basalt fiber.”

Author Response

Thanks so much for your suggestion.

Round 2

Reviewer 3 Report

The points asked were addressed properly.